A genomic perspective on the potential of termite-associated Cellulosimicrobium cellulans MP1 as producer of plant biomass-acting enzymes and exopolysaccharides

Vu Nguyen Thi-Hanh 1 2
Quach Tung Ngoc 1 2
Dao Xuan Thi-Thanh 3 4
Le Ha Thanh ha.lethanh@hust.edu.vn 3
Le Chi Phuong 1
Nguyen Lam Tung 3
Le Lam Tung 1
Ngo Cuong Cao 5
Hoang Ha 1
Chu Ha Hoang 1 2
Phi Quyet-Tien tienpq@ibt.ac.vn 1 2
1 Institute of Biotechnology, Vietnam Academy of Science and Technology, Hanoi, Vietnam , Hanoi , Vietnam
2 Graduate University of Science and Technology, Vietnam Academy of Science and Technology , Hanoi , Vietnam
3 School of Biotechnology and Food Technology, Hanoi University of Science and Technology , Hanoi , Vietnam
4 Vinh University , Vinh , Vietnam
5 Vietnam–Russia Tropical Center , Hanoi , Vietnam
Moyer Craig
Electronic publication date: 2021 Jul 28
Publication date: 2021
Volume: 9
Electronic Location ID: e11839
Received 2021 Mar 11; Accepted 2021 Jul 1
Copyright: ©2021 Vu et al.
Copyright year: 2021
Copyright holder: Vu et al.
License: This is an open access article distributed under the terms of the Creative Commons Attribution License, which permits unrestricted use, distribution, reproduction and adaptation in any medium and for any purpose provided that it is properly attributed. For attribution, the original author(s), title, publication source (PeerJ) and either DOI or URL of the article must be cited.
License URL: https://creativecommons.org/licenses/by/4.0/

Keywords: Carbohydrate-active enzymes, Cellulosimicrobium cellulans, Levan, Lignocellulose, Termite guts, Whole-genome sequencing

Funding: The Institute of Biotechnology, Vietnam Academy of Science and Technology (VAST) CN40SH.01/18-21 This study was financially supported by the Institute of Biotechnology, Vietnam Academy of Science and Technology (VAST) under grant number: CN40SH.01/18-21. The funders had no role in study design, data collection and analysis, decision to publish, or preparation of the manuscript.

==============================
Background

Lignocellulose is a renewable and enormous biomass resource, which can be degraded efficiently by a range of cocktails of carbohydrate-active enzymes secreted by termite gut symbiotic bacteria. There is an urgent need to find enzymes with novel characteristics for improving the conversion processes in the production of lignocellulosic-based products. Although various studies dedicated to the genus Cellulosimicrobium as gut symbiont, genetic potential related to plant biomass-acting enzymes and exopolysaccharides production has been fully untapped to date.

Methods

The cellulolytic bacterial strain MP1 was isolated from termite guts and identified to the species level by phenotypic, phylogenetic, and genomic analysis. To further explore genes related to cellulose and hemicellulose degradation, the draft genome of strain MP1 was obtained by using whole-genome sequencing, assembly, and annotation through the Illumina platform. Lignocellulose degrading enzymes and levan production in the liquid medium were also examined to shed light on bacterial activities.

Results

Among 65 isolates obtained, the strain MP1 was the most efficient cellulase producer with cellulase activity of 0.65 ± 0.02 IU/ml. The whole genome analysis depicted that strain MP1 consists of a circular chromosome that contained 4,580,223 bp with an average GC content of 73.9%. The genome comprises 23 contigs including 67 rRNA genes, three tRNA genes, a single tmRNA gene, and 4,046 protein-coding sequences. In support of the phenotypic identification, the 16S rRNA gene sequence, average nucleotide identity, and whole-genome-based taxonomic analysis demonstrated that the strain MP1 belongs to the species Cellulosimicrobium cellulans. A total of 30 genes related to the degradation of cellulases and hemicellulases were identified in the C. cellulans MP1 genome. Of note, the presence of sacC1-levB-sacC2-ls operon responsible for levan and levan-type fructooligosaccharides biosynthesis was detected in strain MP1 genome, but not with closely related C. cellulans strains, proving this strain to be a potential candidate for further studies. Endoglucanases, exoglucanases, and xylanase were achieved by using cheaply available agro-residues such as rice bran and sugar cane bagasse. The maximum levan production by C. cellulans MP1 was 14.8 ± 1.2 g/l after 20 h of cultivation in media containing 200 g/l sucrose. To the best of our knowledge, the present study is the first genome-based analysis of a Cellulosimicrobium species which focuses on lignocellulosic enzymes and levan biosynthesis, illustrating that the C. cellulans MP1 has a great potential to be an efficient platform for basic research and industrial exploitation.

Introduction

Termites are social insects contributing to nutrient recycling in terrestrial ecosystems and many vertebrate food chains (Wong et al., 2014). Termites utilize microbes in the hindgut to hydrolyze efficiently cellulose of wood and lignocellulosic materials, into more easily digested sugars and short-chain fatty acids (Ohkuma, 2003; Pasari et al., 2019). As demonstrated previously, termites were not able to digest the lignocellulosic biomass, a mixture of cellulose, hemicellulose, lignin, and pectin completely in the absence of symbiotic microorganisms such as bacteria, archaea (Tokuda & Watanabe, 2007; Wong et al., 2014). Thus, termite gut served as a promising source of identifying novel cellulolytic enzymes as well as an excellent model for further investigating the symbiotic relationships between bacteria and their host.

Recently, bacterial symbionts are acquiring much attention from researchers over the world as potential sources for screening novel and highly efficient lignocellulose-degrading enzymes. They can be applied in many industries like biofuel, food, pulp and paper, and agriculture (Chutani & Sharma, 2016; Pandey, Edgard & Negi, 2016). Species belonging to genus Bacillus, Brevibacillus, Cellulomonas, Streptomyces, and Paenibacillus were well-studied as excellent cellulases, hemicellulases, xylanase and pectinases producers (Kamsani et al., 2016; Ohkuma, 2003; Pasari et al., 2019; Ventorino et al., 2016). Cellulose hydrolysis is attributed to the synergistic activity of three different groups such as endoglucanase, exoglucanase, and β-glucosidase (Jäger & Büchs, 2012; Wi et al., 2015). Typical hemicellulases are arabinoxylanases, mannanase, and xylanases involved in hemicellulose decomposition (Dashtban, Schraft & Qin, 2009). The whole-genome sequencing approach was used to promote rapid advances in the discovery of potent cellulase and hemicellulose enzymes. In regard to 20 identified CAZymes including 12 endoglucanases, two exoglucanases, and 6 β-glucosidases, halophilic bacterium Parvularcula flava NH6-79T serves as cellulolytic enzymes producer (Abdul Karim et al., 2020). Micromonospora sp. CP22 was reported to depolymerize the lignocellulosic biomass based on 63 cellulolytic and hemi-cellulolytic CAZymes found in the genome (Chen et al., 2020). The genome B. velezensis LC1 was shown to contain 31 genes involved in lignocellulose degradation, and some of these genes were highly induced in presence of bamboo shoot power (Li et al., 2020). This finding is especially significant given that few genomes of termite gut symbiotic bacteria are available.

Levan has received increasing scientific attention due to its application in the pharmaceutical, industrial, and food fields (Belghith et al., 2012; Dahech et al., 2011; Yoo et al., 2004). Indeed, levan is mainly composed of β-2,6 polyfructan with extensive branching through β-(2,1) linkages, that are mostly synthesized by bacterial enzymes (Gojgic-Cvijovic et al., 2019; Mardo et al., 2017). Japan, the US, and South Korea allowed food and pharmaceutical companies to manufacture levan as a functional food additive (Verspreet et al., 2015), while it is not commercially permitted in Europe (Mardo et al., 2017). Bacterial levan is synthesized in sucrose-rich environments through the action of levansucrase (EC 2.4.1.10) for energy reserve and biofilm formation (Shih et al., 2005). Under starvation conditions, accumulated levan was found to be converted into levan-type fructooligosaccharides (L-FOs) that are imported across the outer membrane (Gray et al., 2021). Until now, many bacteria were reported to produce levan such as Bacillus, Erwinia, Pseudomonas, Microbacterium, and Zymomonas (Gojgic-Cvijovic et al., 2019). The whole genome sequencing shed light on the full potential of levansucrase and levanse in levan-producing bacteria. As revealed in B. subtilis, sacB gene encoding for levansucrase catalyzes the synthesis of levan, which is then degraded mainly into levanbiose by the action of levanase such as YveA and YveB. In addition, the sacB–yveB–yveA levansucrase tricistronic operon is conserved across 12 complete genome sequences of B. subtilis (Dogsa et al., 2013). Interestingly, sacB gene was also found to be conserved in halophilic bacterium Halomonas smyrnensis AAD6R (Diken et al., 2015).

In previous studies, complete genome sequencing showed that genus Cellulosimicrobium is a rich source of glycosidases involved in plant-growth promoting and ginseng biotransformation abilities (Eida et al., 2020; Zheng et al., 2017). To the best of our knowledge, genomic analysis of Cellulosimicrobium has not been revealed to prove a better understanding of its genetic basis for other applications. In this study, we report for the first time, the identification and detailed genomic analysis of cellulose-degrading and levan-producing Cellulosimicrobium cellulans MP1 isolated from the termite gut. These findings provided a scientific basis for the further employment of strain MP1 and its potential genes in biotechnological processes.

Materials & Methods

Isolation of symbiotic cellulolytic bacteria

Drywood termites (Cryptotermes domesticus) were collected from rotten tree trunks and bagasse in Nghe An Province, Vietnam. Termites were surface-sterilized using 70% ethanol to remove contamination and then washed with sterile distilled water. The head of each termite was removed using forceps; the entire guts were removed, crushed with glass rods, and subsequently inoculated into 1 ml broth mineral medium M1 (NaNO3 2.5 g; NaCl 0.1 g; KH2PO4 2 g; MgSO4 0.2 g; CaCl2.6H2O 0.1 g, pH 7.0 in a liter) containing 1% carboxymethylcellulose (CMC) or filter paper as a sole carbon source (Gupta, Samant & Sahu, 2012). These cultures were then incubated for 14 days in an incubating sharker at 30 °C. To isolate the cellulolytic bacteria, the growing cultures were spread on the M1 plate medium (K2HPO4 1 g; NaNO3 2.5 g; KCl 2 g; peptone 2 g; MgSO4 0.5 g; CMC 10 g; agar 15 g; pH 7.0 in a liter). All bacterial isolates were subsequently purified by re-streaking on the M2 agar plate. Confirmation of the cellulolytic ability of pure isolates was performed on the solid medium by covering the Petri dishes with Congo-red dye (Teather & Wood, 1982). The colonies showing yellow-colored halo zones by Congo-red staining were considered as positive cellulolytic bacteria and the clear zones were measured.

Physiological, biochemical, and 16S rRNA sequencing analysis

The shape and size of strain MP1 were determined by scanning electron microscope (SEM) JSM-5410 (JEOL, Tokyo, Japan). Gram staining was performed using the conventional methodology and confirmed using the KOH test (Powers, 1995). The effects of different conditions (ranges of pH, NaCl concentration and temperature, carbon and nitrogen sources) on the growth were investigated as previously described (Kamlage, 1996).

Genomic DNA for 16S rRNA gene sequencing was prepared by phenol-chloroform extraction. The amplification of the 16S rRNA gene sequence of strain MP1 was performed by using the universal primer pair 27F (5′ -TAACACATGCAAGTCGAACG-3′) and 1429R (5′-GGTGTGACGGGCGGTGTGTA-3′) (Phi et al., 2010). A sequence similarity search was carried out using the BLAST program (http://blast.ncbi.nlm.nih.gov/Blast.cgi). The phylogenetic tree was computed by using the neighbor-joining method with 1,000 bootstrap in MEGA version 6.0 (Tamura et al., 2013). Numbers at nodes indicate percentages of 1000 bootstrap re-samplings and Bifidobacterium bifidum DSM 20456 (S83624) was used as the outgroup branch. The 16S rDNA gene sequence of strain MP1 was deposited onto the GenBank (NCBI) under accession number MW534740.

Genome sequencing, de novo assembly, and annotation

For library construction, DNA was extracted from a pure culture of a single colony of strain MP1 using G-spin™ Total DNA Extraction Mini Kit according to the manufacturer’s instructions. The quantity and quality of extracted DNA were evaluated by electrophoresis in 0.6% (w/v) agarose gel and NanoDrop spectrophotometer 2000 Thermo Scientific in order to construct a paired-end library. The constructed genome library was then sequenced using the Illumina platform (Illumina, California, USA). Quality control and read trimming were conducted using FastQC version 0.11.5 (http://www.bioinformatics.babraham.ac.uk/projects/fastqc) and Trimmomatic version 0.36 (Bolger, Lohse & Usadel, 2014). The de novo genome assembly was made with SPAdes v.3.13 (Bankevich et al., 2012), which was then analyzed for its completeness using the Benchmarking Universal Single-Copy Orthologous (BUSCO) version 3 (https://gitlab.com/ezlab/busco).

The draft genome assembled into contigs was annotated using Prokaryotic Genomes Annotation Pipeline (PGAP; http://www.ncbi.nlm.nih.gov/genome/annotation_prok/) at NCBI. In addition, the CRISPRCasFinder was used to identify putative CRISPR loci and Cas cluster (Grissa, Vergnaud & Pourcel, 2007). Orthologous genes and Gene ontology (GO) were analyzed using clusters of orthologous genes (COGs) (Galperin et al., 2015) and InterProScan 5 (Jones et al., 2014), respectively. Virulence factors-encoding genes were identified using the Pathosystems Resource Integration Center (PATRIC) platform (Wattam et al., 2017). The graphical map of the circular genome was also generated using PATRIC. The draft genome sequence was deposited in the GenBank (NCBI) database under accession number: JAFGYF000000000.

Analysis of whole-genome similarity

To classify strain MP1 at the species level, whole-genome similarity including average nucleotide identity (ANI) calculation and digital DNA–DNA hybridization (dDDH) was performed. The ANI was calculated using the orthologous average nucleotide identity (OrthoANI) (Lee et al., 2016). The MP1 genome sequence data was uploaded to the Type (Strain) Genome Server (TYGS) for a whole-genome-based taxonomic analysis (https://tygs.dsmz.de). In silico dDDH and the branch lengths were and the Genome BLAST Distance Phylogeneny evaluated using Genome-to-Genome Distance Calculator (GGDC) (Meier-Kolthoff et al., 2013).

Comparative genomics and prediction of Carbohydrate-active enzyme

A genome-wide comparison of COGs between the assembled genome of Cellulosimicrobium cellulans MP1 and three other C. cellulans available in GenBank, including J36 (NZ_JAGJ01000000.1), LMG16121 (NZ_CAOI01000000.1), and ZKA 48 (NZ_QUMZ01000000.1) was implemented using the OrthoVenn web server with default parameters (E-value 1e−5 and inflation value 1.5) (Wang et al., 2015). The putative genes encoding CAZymes in the comparative genomes were predicted using the dbCAN2 meta server and classified by DIAMOND, HMMER, and Hotpep via CAZy and dbCAN databases, respectively. The top hits with e-value <1E−17, minimum homology rate >50%, and coverage >45% were considered to be homologs.

Determination of extracellular enzymatic activities

To verify the production of endoglucanase, C. cellulans MP1 was cultivated in the TN medium (rice bran 20 g; soya peptone 10 g, casein 10 g, NaCl 1 g, pH 7.0 in a liter) at 37 °C with vigorous sharking. At different time intervals, samples were taken, followed by centrifugation at 4 °C, 12,000 rpm for 10 min to remove bacterial cells and insoluble materials from the culture broth. About 0.5 ml of the crude enzyme solution was added into 0.5 ml of 0.05 M sodium phosphate, pH 7.0 buffer containing 1% of CMC. The mixture was then incubated at 30 °C for 30 min, which was stopped by adding 1 ml of 3,5′–dinitrosalicylic acid (DNS) reagent followed by boiling the reaction mixture at 100 °C for 5 min (Miller, 1959). As for xylanase production, strain MP1 was cultivated on M1 medium (K2HPO4 1 g; KCl 2 g; NH4Cl 2.5 g; yeast extract 2 g, MgSO4 0.5 g, sugar cane bagasse 5 g, pH 7.0 in a liter) at 37 °C for 4 days. At different time intervals, culture was centrifuged at 4 °C, 12,000 rpm for 10 min to obtain the crude enzyme. About 0.5 ml of the crude enzyme was mixed with 0.5 ml of the substrate solution and incubated at 50 °C for 5 min, followed by adding 1 ml DNS reagent. Xylanase activity was assayed using 1% birchwood xylan solution in acetate buffer, pH 5.0 as the substrate (Bailey, Biely & Poutanen, 1992). To quantify pectinase activity, the M2 medium (NH4Cl 2.5 g; K2HPO4 1 g; MgSO4 0.5 g; KCl 2 g; yeast extract 2 g, rice bran 10 g; lactose 10 g; pH 7.0 in a liter) was used. The pectinase assay was performed at 40 °C for 30 min as described previously (Mercimek Takcı& Turkmen, 2016). The quantitative enzyme assays of endoglucanase, xylanase, and pectinase were performed according to standard IUPAC procedures and expressed as international units (IU) (Ghose, 1987). One unit (IU) of enzymatic activity is defined as the amount of enzyme that releases 1 µmol reducing sugars per ml of culture supernatant per minute under assay conditions.

Levan determination

To test the ability to synthesize levan, the C. cellulans MP1 was cultured overnight in batch culture medium (yeast extract 7 g, (NH4)2SO4 1.5 g, K2HPO4 2.5 g, pH 7.0 in a liter) at 37 °C under vigorous agitation. The overnight culture was transferred to a new batch culture medium supplemented with 100 and 200 g/l sucrose and adjusted to an optical density at 600 nm of 0.1. The culture was centrifuged at 10,000 rpm for 10 min at different time intervals. Cell-free supernatant was used to determine levan produced during fermentation (Gojgic-Cvijovic et al., 2019). Levan was harvested by adding three volumes of ice-cold ethanol. The mixture was kept at 4 °C for 12 h, centrifuged at 12,000 rpm at 4 °C for 20 min, and washed with 75% ethanol to remove the residual sugars. The obtained precipitate was hydrolyzed with 0.1 M HCl at 100 °C for 1 h. After neutralizing with 2 M NaOH, levan content was determined according to Somogyi and Nelson (Somogyi, 1945).

Results

Cellulose-degrading potential and identification of the strain MP1

Of the 65 isolates able to grow on CMC plates as the sole carbon source, a total of 8 bacterial isolates produced variable zones of CMC clearance after Congo-red staining. Among those, MP1 was selected due to showing the maximum zone of clearance (9 ± 1, 1 mm). In support of this result, enzyme assays for cellulase activity on CMC and filter paper were found to be the highest for MP1 with 0.66 ± 0.15 IU/ml after 48 h and 0.33 ± 0.05 FPU/ml after 72 h of cultivation, respectively (Table S1). These results indicated that isolate MP1 is a potent cellulolytic bacterium for further study.

The phenotypic examinations indicated that MP1 grew well on LB agar after 2 days of incubation at 37 °C, producing colonies that were circular, smooth, convex, and pale yellow in colour. MP1 cells were Gram-positive, non-spore-forming rods, and nonmotile. The strain MP1 could utilize D-galactose, D-sucrose, D-raffinose, and amygdalin as sole carbon and energy (Table S2). It gave a positive test for catalase, starch hydrolysis, β-galactosidase, and nitrate reduction, whereas negative for oxidase, gelatinase, urease, indole, H2S, acetoin. The 16S rDNA gene sequence of MP1 was aligned with the similar nucleotide sequences in the GenBank database in which a phylogenetic tree was then constructed. The neighbor-joining phylogenetic tree showed the close relationship between MP1 and related Cellulosimicrobium species and the highest homology to Cellulosimicrobium cellulans DSM 43879 (99.5%) and Cellulosimicrobium funkei A153 (99.6%) (Fig. 1). Further, the OrthoANI software was performed to determine the OrthoANI value between isolate MP1 and five closely related Cellulosimicrobium species. It revealed that MP1 shared high similarity to C. cellulans DSM 43879 (88.71%), C. funkei JCM 14302 (91.35%), and C. funkei JCM NRBC 104118 (91.35%), and low nucleotide similarity was observed with Promicromonospora sukumoe SAI-064 that were out of distinct sub-clade (Fig. 2A). This result confirmed that this strain was not considered to be a novel species.

Figure 1 Identification of the strain MP1.

The Maximum-likelihood phylogenetic tree based the 16S rRNA sequence of strain MP1 and representatives of reference type strains. Bar: 0.05 substitutions per site.

Figure 2 Phylogenomic classification of Cellulosimicrobium sp. MP1 based on genome analysis.

(A) Heatmap of OrthoANI values for Cellulosimicrobium sp. MP1 and five closely related species. (B) Genome Basic Local Alignment Search Tool (BLAST) distance phylogenies (GBDP) using Type Strain Genome Server (TYGS) platform. Branch lengths are scaled in terms of GBDP distance formula d5. Numbers above branches are GBDP pseudo-bootstrap support values > 60% from 100 replications, with an average branch support of 84.3%. Tree was rooted at midpoint.

Figure 3 Circular genome map of C. cellulans MP1.

To make identification more accurate at the species level, the whole-genome-based taxonomic analysis conducted by the Type Strain Genome Server (TYGS) platform suggested that Cellulosimicrobium sp. MP1 was closest to C. cellulans DSM 43879 with digital DNA-DNA hybridization (dDDH) values and differences in guanine-cytosine (GC) content of 57.5% (formula d6) and 0.56%, respectively (Fig. 2B). As shown previously, the differences between C. funkei and C. cellulans in phenotypic characteristics are motility and ability to utilize raffinose (Hamada et al., 2016; Yoon et al., 2007). Based on the phenotypic characteristics and genome-wide comparison, this strain was identified as Cellulosimicrobium cellulans MP1. This bacterium was deposited at VAST-Culture Collection of Microorganisms (VCCM) with the accession number VCCM 14150.

Genome sequence and general features of the C. cellulans strain MP1

Briefly, the standard short insert library yielded 443,986,169 bases resulting in 4,487,842 mapped reads with about 92.2-fold sequencing depth. The draft genome of strain MP1 (4,580,223 bp with GC content of 73.9%) was assembled into a ring chromosome, comprising of 23 contigs and no plasmid was detected (Fig. 3). The genome was predicted to have 4,088 genes assigned for 3,964 protein-coding sequences (CDS), 61 rRNA sequences, 55 tRNA sequences, 3 ncRNA sequences (Table 1). Moreover, a total of three virulence factors present in C. cellulans MP1 included dihydroxy-acid dehydratase, CarD-like transcriptional factor, and two calmodulin-like proteins (Fig. 3).

A total of 79.5% (3,216 out of 4,046) of the protein-coding sequences were assigned to 21 out of 25 COG functional categories (Fig. 4A). Transcription (K: 365 protein-coding sequences), carbohydrate transport and metabolism (G: 307), amino acid transport and metabolism (E: 224), energy production and conversion (C: 199), and inorganic ion transport and metabolism (P: 180) were found to be the largest categories. By contrast, the least represent groups included cell cycle control, cell division, chromosome partitioning (D: 33), chromatin structure and dynamics (B: 2), and cell motility (N: 1).

Table 1 Genomic features of C. cellulans MP1.

Features	Chromosome	
Genome size (bp)	4,580,223	
No. of contigs	23	
No. of plasmids	0	
G + C content (%)	73.9	
Genes (total)	4,088	
CDSs (coding)	3,964	
rRNAs	61	
tRNAs	55	
ncRNAs	3	
Pseudogenes	63	
CRISPRS	0	
GenBank accession number	JAFGYF000000000	

Figure 4 The functional annotations of C. cellulans MP1.

(A) Cluster of orthologous gene (COG) classification. (B) Gene ontology (GO) functional classification.

GO analysis was used to provide a deeper understanding of the functional catalogs of strain MP1. A total of 3,165 genes were assigned to 43 subclasses, including 13 subclasses of the cellular component (CC) class, 10 subclasses of the molecular function (MF) class, and 20 subclasses of the biological process (BP) class (Fig. 4B). In detail, the CC class occupied the most genes (1638 genes; 45.3%), followed by the BP (1424 genes; 39.4%), and MF (553 genes; 15.3%) class. The most abundant pathways were cell (GO:0005623; 508 genes), cell part (GO:0044464; 508 genes), and membrane (GO:0016020; 340 genes), which could be considered as the main functional groups of genes belonging to the CC class. Within the BP class, the three most prevalent molecular functions were metabolic process (GO:0008152; 378 genes), cellular process (GO:0009987; 379 genes), and growth (GO:0040007; 281 genes) (Fig. 4B). Among all subclasses belonging to the MF class, catalytic activity (GO:0003824) and binding (GO:0005488) contributed the most genes, with 311 and 176, respectively.

Genome-wide comparison of COGs and Carbohydrate-active enzymes

Given that microbial evolution is due to vertical descent from a single ancestral gene leading to orthologous genes in different species, it is necessary to explore gene function, gene structure, and molecular evolution by using a genome-wide comparison of COGs in different strains. The COGs of C. cellulans MP1 were compared with four other strains, including J36, LMG16121, and ZKA48. It showed that C. cellulans MP1 comprised of 3405 COGs and 794 singletons. C. cellulans J36 and ZKA48 included 3604 and 3605 COGs, respectively, whereas the lowest COGs were observed in LMG16121 (Fig. 5A). Surprisingly, strain MP1 had the largest number of singletons (n = 794). Venn diagram denoted that a total of 2,539 COGs were commonly shared by all four strains of C. cellulans. Among the unique COGs observed in all strains, strain MP1 had the largest number of 20 (Fig. 5A). To support this result, the REALPHY phylogeny builder web tool was used to compare C. cellulans genomes. A maximum-likelihood phylogenetic tree showed that MP1 and LMG16121 formed a monophyletic clade, suggesting a strong evolutionary relationship (Fig. S1).

Figure 5 Comparative genomes between C. cellulans MP1 and three other C. cellulans strains.

(A) Venn diagram represents the numbers of unique and shared orthologous genes of each strain. (B) Comparative genomic analysis of CAZymes across C. cellulans strains.

Using dbCAN 2 meta server, the putative genes encoded for CAZymes present in C. cellulans MP1 were screened to find out the genes responsible for cellulose and hemicellulose degradation. After removing the sequences that did not meet the filtering criteria, a total of 195 predicted CAZymes was identified corresponding to 4.8% of the total of 3,216 protein-coding sequences observed in this strain. Glycoside hydrolases (GHs) involved in the degradation of the most plant biomass such as cellulose and hemicellulose were predicted to be the most abundant subfamily with 99 enzymes. Next, 39 glycosyltransferases (GTs), 37 carbohydrate-binding modules (CBMs), 11 carbohydrate esterases (CEs), 6 enzymes with auxiliary activities (AAs), and 3 polysaccharide lyases (PLs) were detected (Fig. 5B).

The genome of C. cellulans MP1 was compared to closely related C. cellulans such as C. cellulans J36, C. cellulans LMG16121, C. cellulans ZKA48. Generally, C. cellulans MP1 had significantly more CAZyme domains found than other C. cellulans strains. Despite forming a monophyletic clade with strain LMG16121, MP1 was 56 CAZyme domains higher than LMG16121. Meanwhile, with the exception of LMG16121, the family numbers of GT, CBM, and AA family numbers were the same in all compared strains, suggesting the coexistence of these genes in breaking down cellulosic biomass (Fig. 5B).

Mining of plant biomass-acting enzymes in C. cellulans MP1 gene pool

CAZyme analysis suggested that 30 cellulose-related sequences were detected in the genome of C. cellulans MP1. The major families related to the degradation of cellulose are GH6, GH9, GH48, GH10, GH16, GH1, GH3, GH13, and GH64. Based on the annotation, 5 endoglucanases (two GH6 and three GH9), 3 exoglucanases (GH6, GH10, and GH48), and lichenase (GH16) were revealed in the genome of strain MP1 (Table 2). A total of five out of the eight annotated endoglucanases and exoglucanases (orf_454, orf_1616, orf_3244, orf_1607, orf_1611) contained a CBM2 domain appended to them, which have been known to bind to crystalline cellulose, insoluble chitin, and xylan (McLean et al., 2002). CBM2 was the major CBMs present in the genome of strain MP1. In the β-glucosidases family, three GH1 and six GH3 were considered as other important members for cellulose degradation and three GH13 had α-glucosidase activity. It is interesting to note that two pectate lyases such as PL4 (orf_102) and PL1 (orf_1326) belonging to the large class of PL were identified (Table 2). Given that pectin is involved in providing structural support for plant such as cell walls, wall integrity, and cell–cell cohesion, pectate lyases (EC 4.2.2.2) belonging to pectinase catalyze the eliminative cleavage of α-1,4-glycosidic bonds between C4 and C5 of pectin or pectic acid, producing unsaturated methyloligogalacturonates (Abbott, Gilbert & Boraston, 2010; Hugouvieux-Cotte-Pattat, Condemine & Shevchik, 2014). Pectate lyases are important symbiotic bacteria through facilitating their growth in presence of highly pectinolytic bacteria and under pectin-rich environments (Hugouvieux-Cotte-Pattat, Condemine & Shevchik, 2014).

Table 2 List of predicted cellulolytic and hemicellulolytic enzymes present in the genome of C. cellulans MP1.

Classification	Locus tag	Predicted function	
Cellulose-related	Orf_454, Orf_1616, Orf_2130, Orf_2755, Orf_3244	Endoglucanase [EC 3.2.1.4]	
Orf_1607, Orf_1610, Orf_1611	Exoglucanase [EC 3.2.1.91]	
Orf_2289	Lichenase [EC 3.2.1.73]	
Orf_2294, Orf_2388, Orf_2464, Orf_2606, Orf_3385, Orf_401, Orf_403, Orf_404, Orf_922	β-glucosidase [EC 3.2.1.21]	
Orf_2704, Orf_2893	Oligo-1,6-glucosidase [EC 3.2.1.10]	
Orf_2898	Maltodextrin glucosidase [EC 3.2.1.20]	
Orf_802	Glucan endo-1,3- β-glucosidase [EC 3.2.1.39]	
Orf_3703	β-galactosidase [EC 3.2.1.23]	
Orf_2449	Trehalose-6-phosphate hydrolase [EC 3.2.1.93]	
Orf_102	Pectate trisaccharide-lyase [EC 4.2.2.22]	
Orf_1326	Pectate lyase [EC 4.2.2.2]	
Orf_19, Orf_21	Levanase [EC 3.2.1.80]	
Orf_20	Levanbiose-producing levanase [EC 3.2.1.64]	
Orf_22	Levansucrase [EC 2.4.1.10]	
Hemicellulose-related	Orf_905	Mannan endo-1,4- β-mannosidase [EC 3.2.1.78]	
Orf_3988, Orf_4004	Bifunctional β-xylosidase/ α-arabinosidase [EC 3.2.1.37; EC 3.2.1.55]	
Orf_4003	Arabinoxylan arabinofuranohydrolase [EC 3.2.1.55]	
Orf_35, Orf_2605	α-xylosidase [EC 3.2.1.177]	
Orf_24, Orf_2083, Orf_3386, Orf_3999, Orf_4000, Orf_4002	α-L-arabinofuranosidase [EC 3.2.1.55]	
Orf_34, Orf_3650, Orf_4001	Non-reducing end β-L-arabinofuranosidase [EC 3.1.1.185]	
Orf_3698	Exo- α-(1->6)-L-arabinofuranosidase [EC 3.2.1.-]	
Orf_1000, Orf_2117, Orf_3772	Endo-1,4- β-xylanase [EC 3.2.1.8]	
Orf_266, Orf_2247	α-galactosidase [EC 3.2.1.22]	

For hemicellulose degrading genes mining, a total of 21 annotated proteins were deduced to involve in hemicellulose degradation. The major families responsible for the breakdown of hemicellulose were GH43, GH31, GH127, GH51, GH10, GH36, and GH4, and the total number of enzymes, including all families, was 21 (Table 2). Among them, the most abundant enzymes were attributed to GH43 (9 enzymes), followed by GH124 (3 enzymes) and GH31 (2 enzymes), indicating that strain MP1 might have great potentials for degradation of hemicellulosic backbones or debranding hemicellulose. Hemicellulose-related genes were predicted as mannan endo-1,4-β-mannosidase, arabinoxylan arabinofuranohydrolase, α-xylosidase, α-L-arabinofuranosidase, β-L-arabinofuranosidase, endo-1,4-β-xylanase, α-galactosidase, and bifunctional β-xylosidase/ α-arabinosidase (Table 2). Only two enzymes, mannan endo-1,4-β-mannosidase and α-galactosidase, have the ability to hydrolyse parts of mannan, the second major component in hemicellulose. Two endo-1,4-β-xylanase (orf_100 and orf_3772) and one bifunctional β-xylosidase/ α-arabinosidase (orf_3988) were coupled with either CBM2 or CBM9. Surprisingly, genes encoding arabinofuranosidase (orf_3999, orf_4000, orf_4001 and orf_4002), arabinoxylan arabinofuranohydrolase (orf_4003), and bifunctional β-xylosidase/ α-arabinosidase (orf_4004) are clustered in an operon. However, AA enzymes were not found to be related to cellulose and hemicellulose degradation.

To confirm the ability to produce endoglucanase, xylanase, and pectinase of strain MP1, the quantitative enzyme assay was monitored across different incubation periods (24 h, 48 h, 72 h, 96 h, 120 h) (Fig. 6). Using rice bran as an inducible substrate for endoglucanase production, the lowest endoglucanase activity of 0.15 ± 0.01 IU/ml was observed at 24 h, followed by an increase of activity within 48 h to 96 h. The highest endoglucanase production of about 3.17 ± 0.10 IU/ml was achieved after 72 h of incubation. The reduced activity of endoglucanase was notified as 1.40 ± 0.17 IU/ml at 120 h. Xylanase activity of strain MP1 was monitored on M1 medium supplemented with sugar cane bagasse at 37 ° C for 120 h. The result was a steep increase within 0 to 72 h of incubation times. The highest enzyme production was recorded during the stationary phase reaching maximum (1.84 ± 0.08 IU/ml) at 96 h, followed by a slight decrease after 120 h (Fig. 6C). Regarding to pectinase, the maximum pectinase yield was 0.16 ± 0.01 IU/ml after 48 h of incubation. Increasing incubation time was subjected to a significant decrease of pectinase activity. Despite the fact that a number of studies involved in the yields of endoglucanase, xylanase, and pectinase have been performed, it is hard to compare due to the effect of production conditions, substrate and assay conditions, and the way of defining the units.

Figure 6 Enzymatic activities of C. cellulans MP1 observed in different incubation periods.

Mean values with different letters a-d are significantly different according to the Fisher LCD test (P < 0.05).

Levan exopolysaccharide biosynthesis and degradation

Apart from cellulose-related genes, we were interested in the gene encoding for levansucrase (orf_22) since the ability of the genus Cellulosimicrobium to produce levan has not been studied yet. Levansucrase (EC 2.4.1.10) belonging to GH32 participates in synthesizing levan using high sucrose concentration as the main substrate (Raga-Carbajal et al., 2018). The C. cellulans MP1 ls gene consists of 1,851 nucleotides encoding a protein of 616 amino acids with the predicted molecular mass of 66.6 kDa. Alignment of the amino acid sequence of C. cellulans MP1 Ls shows the highest level of identity with Gluconacetobacter diazotrophicus (61%) and Microbacterium saccharophilum (58%) (Fig. S2). In addition, levansucrase is clustered alongside levanbiose-producing levanase (levB-orf_20) and levanases (sacC 1-orf_19 and sacC 2-orf_21) in an operon (Fig. 7A). This result suggested that levan produced by C. cellulans MP1 may be hydrolyzed into L-FOs such as levanbiose and levanases. It is noteworthy that the sacC1-levB-sacC2-ls operon was not present in the compared C. cellulans strains

Figure 7 Levan biosynthesis in C. cellulans MP1.

(A) Genetic organization of levan operon in C. cellulans MP1, Zymomonas mobilis B-14023 and Bacillus subtilis QB112. The ls and sacC-levB homologous genes are denoted in black and gray, respectively. (B) Time course of levan production. (C) Proposed mechanism for levan and levan-type fructooligosaccharide (L-FOs) biosynthesis.

The levan production in media with different initial sucrose concentrations was monitored as shown in Fig. 7B. Levan production depended on cultivation time and sucrose concentration. In the medium with 100 g/l sucrose, the maximum levan concentration was achieved at 20 h (9.9 ± 0.11 g/l), followed by a significant decrease after 24 h (5.5 ± 0.02 g/l). When sucrose concentration was increased to 200 g/l, the highest yield of levan was 14.8 ± 1.2 at 20 h. It was noteworthy that levan was accumulated in the stationary phase.

Discussion

More recently, the exploitation of new plant biomass-acting enzymes and exopolysaccharides with special characteristics has become important since the technological use of agro-industrial residues is increasing. They are exploited favourably in food, paper, cosmetic and pharmaceutical industries (Angelin & Kavitha, 2020; Walia et al., 2017; Zheng et al., 2017). In this context, termites are thought to rely on the gut microbiome to digest wood and other types of plant biomass consisting mainly of cellulose as well as hemicellulose (Calusinska et al., 2020), making them an ideal source to search for new enzymes. Many cellulolytic bacteria from termite have been extensively investigated in the past, despite difficulties in isolation and cultivation (Kamsani et al., 2016). However, despite the use of high-throughput next-generation sequencing, potent biomass-acting enzymes have yet to be revealed. Our work is the first genome analysis systematically elucidating enzymes related to the decomposition of cellulose and hemicellulose and levan production of termite-associated C. cellulans. The findings provide valuable genome information for biotechnological applications.

Despite having potential in many fields, the genus Cellulosimicrobium remains poorly investigated. To date, the genus Cellulosimicrobium consists 7 species (six validly and one non-validly published names): C. variabile (Bakalidou et al., 2002), C. funkei (Brown et al., 2006), C. cellulans (Schumann, Weiss & Stackebrandt, 2001), C. terreum (Yoon et al., 2007), C. marium (Hamada et al., 2016), C. aquatile (Sultanpuram et al., 2015), and C. arenosum (Oh et al., 2018). Only C. variabile was reportedly isolated from the hindgut of termites (Qin et al., 2018; Sharma, Gilbert & Lal, 2016). Since C. cellulans MP1 produced highly active cellulolytic activity on CMC and filter paper, we subsequently analyzed its whole genome for the presence of lignocellulose-degrading CAZymes. Of note, the genome possesses 17 cellulolytic and 21 hemicellulolytic enzymes, in which a total of 11 enzymes featuring CBM domains (CBM2, CBM4, CBM6, CBM9, CBM35) were found. CBM domains play an important role in enhancing the substrate-binding capability of the enzymes, and some CBMs are able to enhance the thermostability of the enzyme (Gilbert, Knox & Boraston, 2013). CBM2 binds to various GH families including GH6 (orf_454, orf_3244), GH9 (orf_1616), GH10 (orf_1000, orf_1611), GH43 (orf_3988), and GH48 (orf_1607), which indicates the ability of these enzymes to bind to and support catalytic domains to hydrolyze crystalline cellulose and xylan. CBM9 domain was found only in endo-1,4- β-xylanase (orf_3772), which is responsible mainly for attacking the xylan backbone and is similar to Clostridium stercorarium Xyn10B and Thermotoga maritima Xyn10A (Lee & Lee, 2014). By contrast, endo-1,4- β-xylanase (orf_2117) had no CBM domain, indicating that other substrate-binding regions might replace the function of CBM. The detection of endoglucanase, exoglucanase, and xylanase activities is in agreement with the presence of the aforementioned genes. It is interesting to note that the strain MP1 cultivated in LB medium containing 0.5% CMC had the lowest endoglucanase but activity of both enzymes increased at least 5.7 fold in TM3 medium supplemented with 2% rice bran, a low-cost agro-residue (Table S3). This result was considered as a promising strategy for reducing the cost of enzyme production and increasing enzyme efficiency.

Pectate trisaccharide-lyase (orf_102) and pectate lyase (orf_1326), which are involved in plant tissue maceration and modification of the cell wall structure (Atanasova et al., 2018), were identified based on the genomic analysis of strain MP1. Due to the ability to cleave pectin using a β-elimination mechanism, pectate lyases are produced either by bacteria living in close proximity with plants or by symbiotic gut bacteria (Biz et al., 2014; Jayani, Saxena & Gupta, 2005). Pectate lyases are important enzymes for industrial applications such as the beverage industry, pulp processing, waste treatment, leading to increasing attention of researchers over the world (Jayani, Saxena & Gupta, 2005; Zhao et al., 2018). Despite 72 genome sequences available at the time of writing and an array of extracellular enzymes found in Cellulosimicrobium, pectate lyases have not been identified yet. C. cellulans MP1 seemed to have acquired 2 different pectate lyases to establish itself in the termite gut. Additionally, pectinase produced by strain MP1 was low activity due to unoptimized medium. The production of pectinase can be optimized by using the response surface methodology as well as the recombinant enzyme, which is an interesting subject for further studies.

Interestingly, the presence of genes related to the levan exopolysaccharide in the genome of C. cellulans MP1 was predictable. Levan is one of the two main types of fructan biopolymers, produced from sucrose through the action of levansucrase (GH68) (Feng et al., 2015). Many studies showed that the GH68 is present in a number of genera, including Gluconobacter, Gluconacetobacter, Komagataeibacter, Asaia, Neoasaia, Bacillus, and Kozakia (Jakob et al., 2019); however, it has not been reported in the genus Cellulosimicrobium. Our finding revealed that the sacC1-levB-sacC2-ls operon is not conserved across well-studied bacteria and that the ls gene is encoded for active levansucrase, catalyzing the synthesis of higher-molecular-weight levan in presence of sucrose, which may serve as carbon storage for C. cellulans MP1. The molecules are hydrolyzed to L-FOs by levan-degrading enzymes including sacC12 and levB (Fig. 7C). Some studies indicate that levanase LevB hydrolyzes levan to generate levanbiose predominantly and SacC is reported to be active against levan and inulin leading to the formation of free fructose (Débarbouillé et al., 1991; Raga-Carbajal et al., 2018). In B. subtilis, anti-terminator, SacY is phosphorylated in presence of the excess sucrose, resulting in upregulation of conserved levansucrase sacB and two endolevanase yveAB (Pereira, Petit-Glatron & Chambert, 2001). Another study demonstrated that activation of sacB also is regulated by pleiotropic regulatory genes degS/degU, degQ and degR (Débarbouillé et al., 1991). While no apparent homologs of SacY have been identified in the chromosome of strain MP1, 4 transcriptional factors degA (orf_114, orf_207, orf_1005, orf_2448) and degU (orf_296, orf_527, orf_1182, orf_2409) are present. Known that deletion of sacC in Zymomonas mobilis enhanced levan production 15.5 g/L to 21.2 g/L (Senthilkumar et al., 2004), further investigation on the regulation mechanisms of this operon is an interesting subject for improving levan production.

Interestingly, Ls shares the 2 conserved cysteines (at positions 383 and 439) and an overall sequence identity of 57–62% with orthologs of gram-negative bacteria such as Microbacterium saccharophilum, Burkholderia vietnamiensis, Gluconacetobacter diazotrophicus (Fig. S2). This result was in contrast to most gram-positive levansucrases that lack a pair of conserved cysteine residues (Jakob et al., 2019). In G. diazotrophicus levansucrase, Cys339-Cys395 intramolecular disulfide bond plays an important role in protein folding and stability. Serin substitution for either Cys339 or Cys395 abolished sucrose hydrolysis activity and levan-forming activity via preventing the extended loop between β-strands IIIB and IIIC with the insertion located between blades III and IV (Martínez-Fleites et al., 2005). Further crystal structure analysis is required to reveal the conformational changes of C. cellulans Ls upon reduction and oxidation conditions.

Conclusions

This study highlighted the ability of strain MP1 to degrade cellulose and hemicellulose and produce levan. Out of 65 isolated termite gut symbiotic bacteria, isolate MP1, identified as C. cellulans, exhibited the highest specific cellulase activity. The genome of C. cellulans MP1 is one of 17 genomes of C. cellulans that are released onto the NCBI genome database, but it is the first sequence that has been reported in detail from a biotechnological perspective. Both genomic and experimental evidence proved that C. cellulans MP1 possesses 30 cellulose and 21 hemicellulose-related sequences, which were functionally redundant for endoglucanases, endoxylanase, β-glucosidases, xylanase, β-xylosidases, arabinofuranosidase, and pectate lyase. Moreover, the sacC1-levB-sacC2-ls operon involved in levan and L-FOs production was pronounced for the first time, which could be a selective advantage during host-adaptation and colonization. These findings not only enrich the genome database but also provide a valuable source of information to continue research into the potential applications of C. cellulans MP1, including its possible use in the biofuel, pulp and paper, and pharmaceutical industries.

Supplemental Information

Supplemental Information 1 Endoglucanase and xylanase of strain MP1 cultivation at interval times

Click here for additional data file.

Supplemental Information 2 Time course of levan production by the strain MP1

Click here for additional data file.

Supplemental Information 3 CMCase and FPase activities of strain MP1 at interval times of cultivation

Click here for additional data file.

Supplemental Information 4 Accession numbers of the 16S rDNA gene and genomic sequences of the strain MP1

Click here for additional data file.

Supplemental Information 5 CMCase and FPase activities of strain MP1 with cultivation times

Click here for additional data file.

Supplemental Information 6 Physiological and biochemical characteristics of strain MP1

Click here for additional data file.

Supplemental Information 7 Predicted genes associated cellulose and hemicellulose degradation

Click here for additional data file.

Supplemental Information 8 Multiple genome alignments were generated by mapping genome sequences of three closely related C. cellulans strains against MP1

Click here for additional data file.

Supplemental Information 9 Protein sequence alignments of levansucrase homologs across known bacteria

Alignment of Ls homologs was performed with Clustal W. Enzyme source and Protein IDs correspond to: Gdia, Gluconacetobacter diazotrophicus (WP_012222901); Bvie, Burkholderia vietnamiensis (WP_011882121); Mce, Cellulosimicrobium cellulans MP1; Msac, Microbacterium saccharophilum (WP_147051238); Achr, Azotobacter chroococcum (WP_052264013). The red and blue color letters indicate percentage of amino acid identity. The conserved Cys is enclosed in the black box and marked with an asterisk (*).

Click here for additional data file.

We are grateful to Emerson Addison, Central Michigan University for critically reading the manuscript.

Additional Information and Declarations

Competing Interests

Author Contributions

Data Availability

The authors declare there are no competing interests.

Nguyen Thi-Hanh Vu conceived and designed the experiments, performed the experiments, prepared figures and/or tables, authored or reviewed drafts of the paper, and approved the final draft.

Tung Ngoc Quach performed the experiments, analyzed the data, prepared figures and/or tables, authored or reviewed drafts of the paper, and approved the final draft.

Xuan Thi-Thanh Dao and Lam Tung Le performed the experiments, prepared figures and/or tables, and approved the final draft.

Ha Thanh Le conceived and designed the experiments, analyzed the data, prepared figures and/or tables, authored or reviewed drafts of the paper, and approved the final draft.

Chi Phuong Le analyzed the data, prepared figures and/or tables, and approved the final draft.

Lam Tung Nguyen, Cuong Cao Ngo and Ha Hoang performed the experiments, analyzed the data, prepared figures and/or tables, and approved the final draft.

Ha Hoang Chu and Quyet-Tien Phi conceived and designed the experiments, analyzed the data, authored or reviewed drafts of the paper, and approved the final draft.

The following information was supplied regarding data availability:

The raw data is available in the Supplemental Files.

The 16S rDNA gene (MW534740) and genomic sequences (JAFGYF000000000) of the strain Cellulosimicrobium cellulans MP1 are available at NCBI GenBank.

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
