# Peer review of "A genomic perspective on the potential of termite-associated Cellulosimicrobium cellulans MP1 as producer of plant biomass-acting enzymes and exopolysaccharides"

_PeerJ, doi:10.7717/peerj.11839_

## Round 0.1 · original submission · Minor Revisions

I agree with all of your reviewers, that minor revisions are needed. I also agree with all three reviewers in assessing that your manuscript is of sufficient quality and merit to warrent publication. Please incorporate each of these suggested corrections and send it back in with the necessary rebuttal letter for a final determination.

Reviewer 1 ·

Basic reporting

The content is clear and supported with relevant references

Experimental design

Well design to justify scope of manuscript

Validity of the findings

The work proposed lacks novelty as similar other strains exist in the scientific domain. A comparative table with outstanding strains can highlight novelty to some extent. So, authors are advised to include a table of comparison.

Additional comments

Please address a few other comments:

1. Authors are advised to explain Paragraph 3 under the results section, clearly and precisely.

2. Figure 4 B could be redesigned (Components not having a significant amount of genes could be excluded).

3. In Figure 6, a decline in enzyme activity after 120 h could be shown.

Reviewer 2 ·

Basic reporting

In this paper, Cellulosimicrobium cellulans MP1 was screened from termite intestine, and genome sequencing and CAZY enzymes analysis were performed; its levan production capacity was also evaluated.The paper is relatively well written and the references are suitable.
There are two main strengths of the thesis.
1. the studied strains were screened from termite gut.
2. the production of levan by Cellulosimicrobium cellulans was reported.
The main disadvantages are:
1. there is not much novelty in the research topic and methodology; especially in the CAZY enzyme analysis part.
2. lack of in-depth study for some interesting research points, for example: sacC1-levB-sacC2-ls operon.

Some specific suggestion:
1. line240-242, the authors found that MP1 has very high homology with DSM 43879 and A153. The current status of research on these highly homologous strains needs to be described. Also, why were these highly homologous strains not used in the subsequent comparative studies?
2. In the chapter "Mining of plant biomass-acting enzymes in C. cellulans MP1 gene pool", please expand the description of PL enzymes and add information about AA enzymes. The authors tested the viability of endoglucanase and Xylanase, did they try the enzyme viability of PL or AA? This experiment may be considered to be added.
3. In Line362-365, the authors suggest that "levan produced by C. cellulans MP1 may be hydrolyzed into L-FOs such as levanbiose and levanases ". In the follow-up experiment, the product assay can be performed to confirm this.

Experimental design

The research question studied and the experimental methods used in this paper are conventional, so there is no problem in the experimental design and methods.This of course makes the novelty of the paper not very strong.

Validity of the findings

no comment

Additional comments

no comment

·

Basic reporting

The originality of this paper appears on reporting for the first time, the identification and detailed genomic analysis of cellulose-degrading and levan-producing among Cellulosimicrobium genus. This paper is added to the few studies combining DNA based approaches to culture-dependent approaches cellulose degradation enzymes in well known and interesting species involved in cellulose biomass degradation.The work’s strengths appear in paper well written, Is the text clear and easy to read. The validity of questions, the use of a detailed methodology and the data analysis being done systematically.

Experimental design

Methodology well described, only some statistical analysis were missing to support your findings.

Validity of the findings

First work to describe and report for the first time, the identification and detailed genomic analysis of cellulose-degrading and levan-producing among Cellulosimicrobium genus.
The Presence of high Cazymes numbers of GH, GT, AA family suggesting the interesting aspect of strain MP1 in breaking down cellulosic biomass.
The Combination of Molecular and cultural approaches to support the findings suggestion the strain MP1 as a good biomass degrading enzymes and Levan producer.

Additional comments

Good introduction that placed the subject in its theorical framework and illustrating the problematic of research.
 The goal of research is clearly reported, regarding isolation of strain MP1 and highlight its role in cellulose degradation and Levan production through a genomic full genome sequencing/analysis.
 Some suggested modifications:
o Line 91: please add a brief definition of what is levan?


 Methodology well described.
 Line 117: could you explain why you make surface sterilization with ethanol?

 Results/discussion sections: Nice findings are here, especially revealed by DNA/metagenome approaches. The combination of both cultural, biochemical and molecular approaches to give more insight on the interesting cellulase of strain MP1for both cellulose degradation and Levan production.
I have some remarks to report below:
 Line 224: remove “isolated from termite gut”, we clear know it from title, introduction and by material/methods.
 Line 230: in table S1, how do you explain the decrease of enzymatic activities after 72H?
 Line 235: how it can be rods and cocci at the same time?
 Based on 16S, genomic analysis, GC%, echophysiology analysis, could you confirm/or not that strain MP1 may represent a new species among Cellulosimicrobium genus?
 In line 246 you reported GC content is 57.5% and in line 258 you said that GC% is 73.9%. which one is correct?
 Figure 3: please remove the text red underline of Cellulosimicrobium cellulans. It would be more talking if you add a color legend of each color in this this figure instead having it figure legend captions.
 I suggest citing Fig. 4 in line 264 instead of line 267.
 Line 296, Could you please add more information in you Figure S1 (similarities %, bootstrapping nodes…)
 Line 388: Cellulosimicrobium consist of 7 species (6 validly published, and one non-validly published):
a. Cellulosimicrobium variabile Bakalidou et al. 2002
b. Cellulosimicrobium terreum Yoon et al. 2007
c. Cellulosimicrobium marinum Hamada et al. 2016
d. Cellulosimicrobium funkei Brown et al. 2006
e. Cellulosimicrobium cellulans (Metcalfe and Brown 1957) Schumann et al. 2001
f. Cellulosimicrobium aquatile Sultanpuram et al. 2016
g. and "Cellulosimicrobium arenosum" Oh et al. 2018

- Missing of statistical analysis.

---

## Round 0.2 · accepted · Accept

I agree with the reviewers in the determination that this manuscript is ready for publication and thank the authors for their attention to all the suggested comments.

Reviewer 2 ·

Basic reporting

Some of the suggested additional experiments were not done, but the authors have basically completed their response to my comments and I think the manuscript is ready to be accepted. The expectation is that they will carry out the experiments I suggested in the future.

Experimental design

no comment

Validity of the findings

no comment

Additional comments

Some of the suggested additional experiments were not done, but the authors have basically completed their response to my comments and I think the manuscript is ready to be accepted. The expectation is that they will carry out the experiments I suggested in the future.

·

Basic reporting

The originality of this paper appears on reporting for the first time, the identification and detailed genomic analysis of cellulose-degrading and levan-producing among Cellulosimicrobium genus. This paper is added to the few studies combining DNA based approaches to culture-dependent approaches cellulose degradation enzymes in well known and interesting species involved in cellulose biomass degradation.

The work’s strengths appear in paper well written, Is the text clear and easy to read. The validity of questions, the use of a detailed methodology and the data analysis being done systematically.

Experimental design

Methodology well described

Validity of the findings

Good findings were reported, especially revealed by DNA/metagenome approaches. The combination of both cultural, biochemical and molecular approaches to give more insight on the interesting cellulase of strain MP1for both cellulose degradation and Levan production.
The study is the first work to describe and report for the first time, the identification and detailed genomic analysis of cellulose-degrading and levan-producing among Cellulosimicrobium genus through combination of Molecular and cultural approaches to support the findings suggestion the strain MP1 as a good biomass degrading enzymes and Levan producer.. The study reported the presence of high Cazymes numbers of GH, GT, AA family suggested the interesting aspect of strain MP1 in breaking down cellulosic biomass.